# DesCo: Learning Object Recognition with Rich Language Descriptions

**Liunian Harold Li**[*]    **Zi-Yi Dou**[*]    **Nanyun Peng**    **Kai-Wei Chang**
University of California, Los Angeles
{liunian.harold.li,zdou,violetpeng,kwchang}@cs.ucla.edu

## Abstract

Recent development in vision-language approaches has instigated a paradigm shift in learning visual recognition models from language supervision. These approaches align objects with language queries (e.g. "a photo of a cat") and thus improve the models' adaptability to novel objects and domains. Recent studies have attempted to query these models with complex language expressions that include specifications of fine-grained details, such as colors, shapes, and relations. However, simply incorporating language descriptions into queries does not guarantee accurate interpretation by the models. In fact, our experiments show that GLIP, a state-of-the-art vision-language model for object detection, often disregards contextual information in the language descriptions and instead relies heavily on detecting objects solely by their names. To tackle the challenge, we propose a new description-conditioned (DesCo) paradigm of learning object recognition models with rich language descriptions consisting of two innovations: 1) we employ a large language model as a commonsense knowledge engine to generate rich language descriptions of objects; 2) we design context-sensitive queries to improve the model's ability in deciphering intricate nuances embedded within descriptions and enforce the model to focus on context rather than object names alone. On two novel object detection benchmarks, LVIS and OminiLabel, under the zero-shot detection setting, our approach achieves 34.8 APr minival (+9.1) and 29.3 AP (+3.6), respectively, surpassing the prior state-of-the-art models, GLIP and FIBER, by a large margin.

## 1 Introduction

Training visual recognition models to classify or detect objects with a fixed set of pre-defined categories has been the convention for a long time. However, models trained using this approach often encounter difficulties when adapting to unfamiliar concepts and domains. Recently, there has been a paradigm shift towards *training visual recognition models with language supervision*, using a contrastive objective on a large amount of image-text data containing a diverse range of visual concepts. These models can then be transferred to downstream tasks via language queries. For example, CLIP [36] can perform image classification using a template query such as "a photo of {class name}"; GLIP [25] can perform object detection by querying the model with "Detect: person, cat, dog $\cdots$".

Early applications of these models typically utilize simple language queries that consist of object names. However, language queries can convey much richer and more comprehensive information, such as object attributes, shapes, textures, and relations. These pieces of information can be especially useful for identifying novel visual concepts that do not appear in the training corpus or specifying

---

[*]Equal contribution.

37th Conference on Neural Information Processing Systems (NeurIPS 2023).

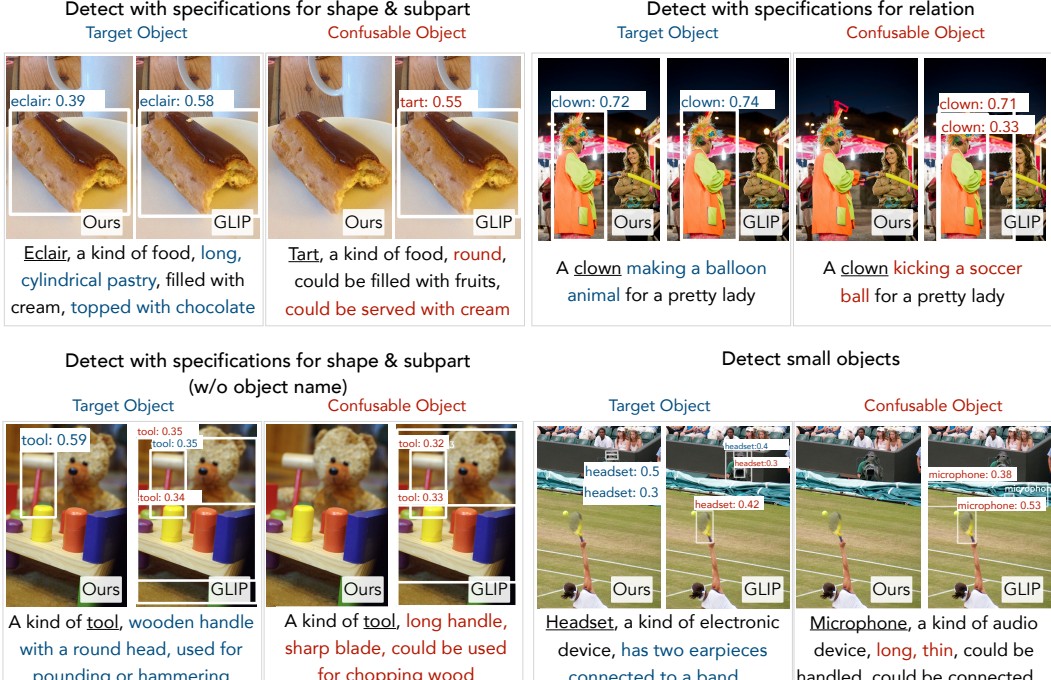

Figure 1: Comparison between our model (DESCO-GLIP) and the baseline (GLIP [25]). Each image is paired with a positive query for target object and a negative query for confusable object. A successful model should locate the target object and ignore the confusable object in the image based on fine-grained specifications for shapes, subparts, relations, etc. We highlight the descriptions that match and do not match to the queried object in blue and red, respectively. Results show that our model can successfully localize the target object and suppresses the negative query even for the difficult cases when the object name is not in the query or the object.

specific needs. For example, the concept of "mallet" can be described as "a kind of tool, wooden handle with a round head" (Figure 1, bottom-left). This decomposes object recognition into recognizing fine-grained details (such as attributes, sub-parts, shapes, etc.) and aligning them to the descriptions. Several studies [25, 41, 31] have explored the idea of guiding language-based recognition models using such descriptive prompts. However, few existing models take complex queries into account during training. As a result, current models often struggle with recognizing intricate object names, attributes, and relations described in natural language sentences [53, 43] (see examples in Figure 1).

In this paper, we develop a vision-language model capable of leveraging description-rich language queries to perform object detection. This work aligns with the recent surge of interest in **instruction/prompt-aware** vision-language models (see a discussion in Section 2). Our goal is to equip VLMs with the ability to comprehend complex language input describing visual concepts, similar to the capability of large language models. We specifically study instruction/prompts in the form of descriptive queries. We focus on object detection, as it requires fine-grained recognition and is more challenging than image-level tasks. However, our method can be generalized to other vision tasks such as classification and segmentation [61].

We identify two major challenges that prevent existing models from efficiently utilizing rich language descriptions: (1) Fine-grained descriptions are rare in image-caption data used for training current VLMs[2]. This resembles the reporting bias phenomenon [34]: when writing captions for images, humans tend to directly mention the entities rather than give a detailed description. (2) Even when provided with data rich in descriptions, models often lack the incentive to leverage these descriptions effectively. The main training objective is to align positive phrases with relevant regions while suppressing negative phrases. However, if positive/negative phrases can be distinguished without

---

[2]We count the region-label data used by models like GLIP as image-caption data because the labels are converted into captions through templates.

descriptions, the training mechanism fails to incentivize the model to use the provided description. For example, a positive phrase like "A toy bear holding a *mallet*, which has a wooden handle with a round head," and a negative phrase like "A toy bear holding an *ax*, which has a long handle and a sharp blade," can be differentiated based solely on the words *mallet* and *ax*. This issue resembles the issue discovered by [53], where vision-language models ignore word order and treat a query as a "bag-of-words" due to insufficient incentives from the contrastive objective. In addition, current models suffer severe hallucination when given natural language queries (in contrast to "template-like" queries) due to shortcuts introduced in training query formulation. This can be seen in the bottom-right picture of Figure 1, where GLIP hallucinates and predicts multiple wrong boxes for "microphone" while "microphone" does not appear in the image.

Based on the observations, we present a **Des**cription-**Co**nditioned (**DESCO**) paradigm of **learning object recognition models from language descriptions** based on two synergistic ideas:

(1) **Generating descriptions with large language models.** Instead of learning from raw image-caption pairs, we use large language models as a world knowledge base and generate detailed descriptions based on the original caption. We prompt GPT-3 [2] with "What features should object detection models focus on for {an entity in the caption}?". This serves as a scalable approach to transfer the image-caption data into image-description data.

(2) **Context-sensitive query construction.** As discussed, even if we provide descriptions during pre-training, models can still ignore the language context. Our solution is to create a "context-sensitive query", which is a set of positive and negative phrases that can only be distinguished by reading the descriptions (Figure 2). We explore two strategies: 1) constructing "Winograd-like" [13, 43] queries by using large language models to generate confusable object descriptions and captions and 2) generalizing the original grounding task to allow full-negative queries, reducing hallucination.

We apply our approach to fine-tune two state-of-the-art language-conditioned object detection models GLIP [25] and FIBER [7]. We use the same raw training data as the baselines but convert the data into description-rich queries. We evaluate our methods in two settings. (1) Zero-shot generalization to novel categories (LVIS [11]), where we use GPT-3 to generate descriptions given class names. DESCO-GLIP (Tiny) improves upon GLIP (Tiny) by 10.0 APr, even outperforming the larger GLIP (Large); DESCO-FIBER improves upon FIBER by 9.1 APr. (2) Zero-shot generalization to natural descriptions given by humans (OmniLabel [38]). DESCO-GLIP and DESCO-FIBER improve upon the baselines by 4.5 AP and 3.6 AP, setting a new state-of-the-art performance level. Code is available at `https://github.com/liunian-harold-li/DesCo`.

## 2 Related work

**Language-based visual recognition models.** Visual recognition models are typically trained to make predictions based on a fixed set of classes [20, 5, 26, 39, 33, 58]. The trained models are hard to generalize to open-domain settings. Recent studies have developed visual recognition models that take into account language queries, i.e. language-based recognition. This line of research can be traced back to early work of generalizing image classification [42] and object detection [1] models with word embeddings. Recently, CLIP [36] reformulates image classification as image-text matching and pre-trains models on large-scale image-caption pairs to learn transferrable representations. They demonstrate strong zero-shot performance on various classification tasks. Recent work has applied the technique to fine-grained recognition tasks, such as object detection [17, 10, 25, 57, 56, 3, 32, 7, 28], and segmentation [21, 9, 16, 47, 54, 24]. These works either use pure image-text data as supervision [47], or reformulate labeled data into image-text data [21], or pseudo labels image-text data with fine-grained labels [25]. Orthogonal to architecture design or scaling-up, which is the focus of many prior studies, this paper points out that the vanilla way of using image-text data is insufficient and studies how to train these models to take more flexible and informative language queries.

**Vision-language models with complex prompts.** As vision recognition models become language-aware and language models become vision-aware [45, 60, 22], there is a growing interest in studying whether these models can take complex language prompts, such as task instructions (e.g., GPV [12, 18], SEEM [62], VisionLLM [46]), descriptions [23], or even dialogues (e.g., LLaVa [27]). We study specifically descriptive prompts, which are especially useful for generalizing to novel categories and customized detection needs; a model that can understand descriptive prompts can also serve as the

backbone for supporting aforementioned other types of prompts. Similar to our work, K-LITE [41] proposes to retrieve knowledge for a visual concept using external knowledge bases, then use the enriched concepts for image classification or object detection; similar techniques have also been proposed by [31, 49, 48]. DetCLIP [50] builds a large-scale concept dictionary, based on which they provide definitions from WordNet. Different from these studies, our methods show that simply presenting the descriptions at training or inference time is not enough; we propose a simple technique to force models to focus on the provided descriptions (Section 3.2.2). Our work relates to a line of work that seek to reveal and fix failure patterns of image-text matching models (e.g., CLIP) by creating hard negative examples [43, 53, 8, 37].

# 3 Approach

In this section, we first briefly introduce language-based object detection models, then illustrate the details of our proposed approach.

## 3.1 Background

We give an introduction to *language-based* object detection models [17, 25, 7], which take a language query and an image as inputs, and predict bounding boxes and their alignment to phrases in the language query. In the following, we use GLIP as an example.

**Baseline: Grounded Language-Image Pre-training (GLIP).**   At the center of these approaches is "reformulating any task-specific fixed-vocab classification problem as a task-agnostic open-vocabulary vision-language matching problem" [55]. The best example is CLIP which reformulates image classification as image-text matching. Similarly, GLIP unifies training data into a *grounding* format: $(I, Q, B, T)$. $I$ is the image; $Q$ is the text query; $B \in R^{N \times 4}$ is the bounding boxes; $T \in \{0, 1\}^{N \times K}$ indicates the ground-truth alignment label between the $N$ bounding boxes and $K$ tokens in the query. The key is how to formulate the *query* with data from two kinds of sources:

- *Detection data.* For object detection data such as Objects365 [39], the query is the concatenation as a list of object classes, such as "Detect: person. bicycle. car. $\cdots$, toothbrush". Note that negative object classes are included in the query; this makes such query-based detection models similar to classical detection models.
- *Grounding data.* Typically, $Q$ is an image caption, containing entities that can are aligned to annotated object regions [35]. For example, "A toy bear holding a mallet" is the caption; "toy bear" and "mallet" are the "groundable" entities. For densely annotated grounding data (multiple captions for one image) [19], we can concatenate multiple captions into a longer query. Image-caption data (without annotated boxes) can be transferred into grounding data via pseudo labeling with a grounding model [25].

Given $I$ and $Q$, we compute the alignment scores $S_{\text{ground}}$ between image regions and words in the query:

$$O, L = \text{Enc}(I, Q), \ S_{\text{ground}} = OL^{\top}, \mathcal{L} = \ loss(S_{\text{ground}}, T) + \mathcal{L}_{\text{loc}}$$

where $L \in \mathbb{R}^{K \times d}$ is the contextual token features and $O \in \mathbb{R}^{N' \times d}$ are the regions features. ENC is a vision and language encoder that takes both image and text as inputs and fuses their representations. The training loss contains the region-word matching loss and a localization loss $\mathcal{L}_{\text{loc}}$)as in conventional object detection models.

**Inference with language query.**   At inference time, the model can be used to locate entities/class names appearing in the query. One could simply provide a list of candidate object names (as in the detection data training format). [25] also show the promise of using descriptions for generalization to novel concepts; however, we show that while GLIP can be influenced by the description, it does not always take the details in the description into account.

## 3.2 Learning with language descriptions

To train object recognition models that fully utilize language descriptions, we propose to generate descriptions with large language models and construct context-sensitive queries during training. The following subsections provide further details.

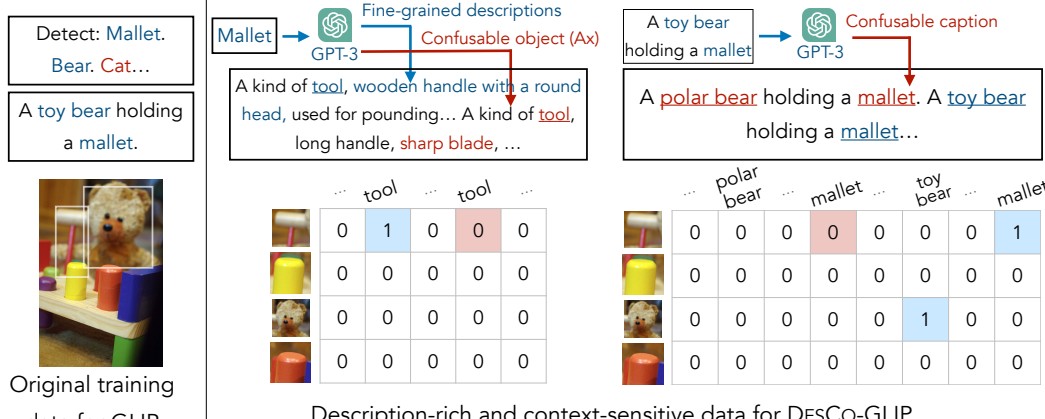

Figure 2: Given the original training data of GLIP, we transform it to be description-rich and context-sensitive by: 1) generating descriptions for entities and composing each of them with confusable object descriptions; 2) generating negative captions. We visualize the gold alignment labels (ground truth) between tokens and regions for the new data. Notably, words such as *tools* are assigned both positive (blue block) and negative (red block) labels in alignment with the corresponding object depending on the context of the query. As such, the model requires understanding the description in order to make the correct prediction.

### 3.2.1 Description generation with large language models

Fine-grained descriptions could be scarce in image-caption data due to reporting bias. While this problem can be alleviated by scaling up the pre-training corpus, we show that large language models [6, 2] can be used to effectively generate the descriptions and thus enrich our training corpus.

We leverage a large language model to transform a query $Q$ into a description-rich query $\text{LLM}(Q)$. In this work, we only focus on generating descriptions for entities mentioned in the original query. We construct a vocabulary consisting of 10K entities appearing frequently in the pre-training corpus. For each entity, we prompt a large language model: `what features should object detection models focus on for {entity}?` We find that large language models give high-quality responses (see examples in Figure 1 and Figure 4). The specific prompts are included in the appendix.

### 3.2.2 Context-sensitive query construction

Can we simply add the description-rich data to the pre-training corpus? An intuitive idea is to append the description to the original entity to form a training prompt (e.g., "Detect: mallet. bear··· " → "Detect: mallet, a kind of tool, wooden handle ··· bear, a kind of animal, ··· "). However, we find that models naively trained with these prompts still do not exhibit "context-sensitivity", i.e., they make predictions solely based on the entity names while ignoring other contexts (see Section 4.1 for quantitative analysis). As a result, we observe no evident benefit in incorporating descriptions during inference (Table 3). In the following, we elaborate on why the model learns to ignore the descriptions and propose two solutions.

**Model learn statistical shortcuts.** We first illustrate that without careful design, the model could learn two statistical shortcuts that make them insensitive to descriptive queries.

(1) *Entity shortcut.* The model is trained to align the entities in the query to image regions (this includes predicting "no-alignment" for entities not appearing in the image). Intuitively, if the alignment can be predicted without relying on the context information in the query, then the model is not incentivized to focus on the context information. Figure 2 illustrates this issue with an example. The left side shows the training data of GLIP, where the top query ("Detect: Mallet. Bear. Cat...") comes from detection data and the bottom query ("A toy bear holding a mallet.") comes from grounding data. The problem with such queries is that they can be grounded by only focusing on the entity names and ignoring the context. We denote the gold alignment label of regions as $T$, the entities in the query as $E$, and the non-entity part (context) of the query as $C$. The mutual information

$I(T; C|E, I)$ between $C$ and $T$ given $E$ and the image $I$ would be low. That is, the non-entity parts of the queries do not affect the label of the region. Training models on such data will not encourage the model to focus on the descriptions as they provide no additional information. This is similar to the "memorization overfit" issue observed in [51]: the model can simply choose to "memorize" the alignment between the entities and regions.

(2) *Format shortcut (hallucination).* Popularized by GLIP [25], a line of work adopts a unified view of phrase grounding and object detection: detection can be seen as language-context-free grounding while grounding can be seen as language-context-dependent detection. However, this unification is still *imperfect*: phrase grounding (or referring expression [52]) traditionally only concerns locating entities in a caption that always exist in the image; thus the model learns to always treat the natural-language queries (in contrast to the template-like queries) as positive and tries to ground every entity mentioned in the sentence. This will result in failure examples as illustrated in the bottom-right picture of Figure 1. Such "hallucination" can be commonly seen on models trained on language grounding data [17]; these models are almost incapable of distinguishing positive and negative "natural-language-like" queries.

**Constructing context-sensitive queries.** This motivates our solution of creating queries that are hard to solve without context (Figure 2 and Figure 3). We explore two strategies in this study.

(1) We construct training queries similar to the Winograd format. For example, when training on detection data, instead of "Detect: mallet, a kind of tool, $\cdots$", we remove the entity name "mallet" from the query and sample another description of a "confusable" entity that is also a kind of tool. Pairing the descriptions of the two "confusable" entities creates a strong supervision signal (the middle example in Figure 2): the alignment label (0 or 1) of the word "tool" now depends on its context. The confusable entities are obtained by prompting the large language models as well. Similarly, for training on grounding data, we prompt language models to generate confusable (hard negative) captions that differ from the original captions only by a few words (the example on the right in Figure 2). Note that the label of the word "mallet" is now affected by the context: the *first* "mallet" is assigned 0 as the caption ("A polar bear holding a mallet") is negative. Mixing in such hard negative captions encourages the model to focus on the context surrounding the entities, such as relations to other entities. To make the confusable caption generation process scalable for image-caption data, we first perform in-context inference and prompt GPT-3 to generate around 50K negative captions based on positive captions; then we distill this knowledge to the open-sourced LLaMA-7B [44] model that is instruction-finetuned using low-rank adaptation[3] [14] and perform inference on large-scale image-caption data.

(2) To resolve the hallucination issue, we generalize the original grounding task: instead of always feeding the model a query that contains at least one description/caption matching the image, we allow the query contain only negative descriptions/captions (Figure 3). Thus, the model cannot blindly ground all entities mentioned in the query; implicitly, it needs to perform image-text matching [36] as well as phrase grounding. This was partly done in GLIP (see Appendix C.1 in the original GLIP paper), but the query still contains at least one positive entity.

**Overview.** In Figure 3, we summarize the overall data construction algorithm. **Algorithm 1**: $B \in R^{N \times 4}$ are the bounding boxes of an image; $E$ are $M$ positive objects (entities) that appear in the image; $V$ are the descriptions of all candidate objects in a pre-defined vocabulary; $T \in \{0, 1\}^{N \times M}$ denotes the gold alignment between boxes and entities. We first prompt LLM to generate descriptions for the positive entities and propose confusable entities and descriptions; the prompt is included in the appendix (Line 3). The original entities are removed from the descriptions with $p_{\text{drop}} = 0.5$ (DropEntity, Line 4-5). Random negative descriptions from the vocabulary are added to the candidate description set (Line 7). We then randomly subsample the descriptions and concatenate them to form a final query $q^*$; this is because the total length of all the candidate descriptions is too large (Line 8). Boxes and the mapping relations between boxes and tokens are accordingly adjusted (Line 9). **Algorithm 2**: $C$ is the original caption and $E$ are $M$ positive phrases we extracted from the caption. The last two lines of both algorithms are important: after *SubSampleConcat*, it is very likely that some positive sub-queries are dropped from $Q$; then *LabelAssign* would drop boxes that are mapped to the dropped sub-queries. The output $B$ could end with fewer boxes or even no boxes. This is

---

[3]https://github.com/tloen/alpaca-lora

**Algorithm 1** Generating Queries for *Detection* Data

**Input:** $B$ (boxes), $T$ (alignment matrix), $E$ (positive entities), $V$ (vocabulary)

1: $Q \leftarrow \emptyset$
2: **for** $i \leftarrow 1$ to $M$ **do**
3:     $q, Q^- \leftarrow$ LLM(prompt, $E_i$)
4:     **if** random() $< p_{\text{drop}}$ **then**
5:         $q, Q^- \leftarrow$ DropEntity($q, Q^-$)
6:     $Q \leftarrow Q \cup \{q\} \cup Q^-$
7: $Q \leftarrow Q \cup$ RandSample($V$)
8: $q^* \leftarrow$ SubSampleConcat($Q$)
9: $q^*, T, B \leftarrow$ LabelAssign($q^*, T, E, B$)

**Algorithm 2** Generating Queries for *Grounding* Data

**Input:** $B$ (boxes), $T$ (alignment matrix), $E$ (positive entities), $V$ (vocabulary), $C$ (caption)

1: **if** random() $< p_{\text{des}}$ **then**
2:     $Q, T, B \leftarrow$ Algorithm1($B, T, E, V$)
3: **else**
4:     $Q^- \leftarrow$ LLM($\text{prompt}_{\text{neg}}, C$)
5:     $Q \leftarrow \{C\} \cup Q^-$
6:     $q^* \leftarrow$ SubSampleConcat($Q$)
7:     $q^*, T, B \leftarrow$ LabelAssign($q^*, T, C, B$)

Figure 3: Algorithms for generating queries from detection data and grounding data.

different from the strategy in GLIP or traditional object detection training recipe, where we strive to keep all boxes provided.

## 4 Experiment

In this section, we first investigate whether current models (GLIP) can utilize language descriptions out-of-the-box; then we show that our method allows the model to utilize language descriptions and improves performance on LVIS and OmniLabel significantly.

### 4.1 Can language-conditioned detection models utilize language descriptions?

As a proof of concept, we first show the GLIP struggles to utilize language descriptions out of the box and analyze the failure patterns.

| Model | $\Delta$Box | $\Delta$Conf | AP |
|---|---|---|---|
| GLIP [25] | 0.291 | 0.05 | 4.7 |
| DESCO-GLIP | **0.381** | **0.11** | **12.4** |

Table 1: GLIP is insensitive to context changes compared to DESCO-GLIP.

**GLIP does not effectively utilize language descriptions.** We make an attempt at using descriptions to transfer GLIP to LVIS [11], which contains over 1,200 classes. The process is similar to that of [31]. For each category, we prompt a large language model (GPT-3) to give details descriptions (as in Section 3) We append the description to the original class name to form a new query. An example of the queries can be seen shown in Figure 1 (bottom row). Directly appending the description to the object name at inference time only degrades the performance: GLIP-T achieves 20.8 AP on rare categories while appending the descriptions makes the performance drop to 12.2 AP. This is likely due to model hallucination on natural-language-like queries.

**GLIP is insensitive to context changes.** Examining the model predictions, we find that the model not only does not utilize language descriptions; it ignores the descriptions and tends to only focus on entity names, as we hypothesized. To quantitatively verify the phenomenon, we introduce a *context-sensitivity* test, inspired by the WinoGround [43] benchmark. For each image, we provide the model with a positive query $q^+$ describing an object that appears in the image and a negative query $q^-$ describing a confusable object. The original object names are removed from the query. An example of the test is shown in Figure 1 (bottom left), where the model is challenged to distinguish "mallet" and "ax". $q^+$ and $q^-$ describe objects from the same general category (e.g., both are "a kind of tool") while differing in other aspects, similar to the Winograd test.

Intuitively, if a model can effectively utilize the descriptions, it should exhibit two properties: 1) it should give higher alignment scores to entities in $q^+$ compared to $q^-$; 2) even if the model cannot "guess" the hidden entity, at least, the model predictions should change drastically when given two different descriptions. We thus introduce two metrics. 1) AP, which measures how accurate

| Model | Backbone | LVIS MiniVal [17] | | | | OmniLabel [38] | | | |
|---|---|---|---|---|---|---|---|---|---|
| | | APr | APc | APf | AP | AP | APc | APd | APd-P |
| MDETR [17] | RN101 | 20.9 | 24.9 | 24.3 | 24.2 | - | - | 4.7 | 9.1 |
| MaskRCNN [17] | RN101 | 26.3 | 34.0 | 33.9 | 33.3 | - | - | - | - |
| RegionCLIP [57] | ResNet-50 | - | - | - | - | 2.7 | 2.7 | 2.6 | 3.2 |
| Detic [59] | Swin-B | - | - | - | - | 8.0 | 15.6 | 5.4 | 8.0 |
| K-LITE [41] | Swin-T | 14.8 | 18.6 | 24.8 | 21.3 | - | - | - | - |
| GroundingDINO-T [28] | Swin-T | 18.1 | 23.3 | 32.7 | 27.4 | - | - | - | - |
| GroundingDINO-L [28] | Swin-L | 22.2 | 30.7 | 38.8 | 33.9 | - | - | - | - |
| GLIP-L [25] | Swin-L | 28.2 | 34.3 | 41.5 | 37.3 | 25.8 | 32.9 | 21.2 | 33.2 |
| GLIP-T [25] | Swin-T | 20.8 | 21.4 | 31.0 | 26.0 | 19.3 | 23.6 | 16.4 | 25.8 |
| DESCO-GLIP | Swin-T | **30.8** | **30.5** | **39.0** | **34.6** | **23.8** | **27.4** | **21.0** | **30.4** |
| FIBER-B [7] | Swin-B | 25.7 | 29.0 | 39.5 | 33.8 | 25.7 | 30.3 | 22.3 | 34.8 |
| DESCO-FIBER | Swin-B | **34.8** | **35.5** | **43.9** | **39.5** | **29.3** | **31.6** | **27.3** | **37.7** |

Table 2: Zero-shot transfer to LVIS and OmniLabel. Numbers that are grayed out are supervised models. DESCO-GLIP and GLIP-T are directly comparable; DESCO-FIBER and FIBER-B are directly comparable; the rest are listed for reference and not directly comparable.

the model's predictions are. 2) $\Delta$Box and $\Delta$Conf, which are the differences between the model's predictions for $q^+$ and $q^-$. $\Delta$Box measures the changes in box coordinates while $\Delta$Conf measures the changes in alignment scores of boxes. Details of the metrics are in the appendix.

We find that the baseline model not only cannot identify the correct description (low AP); but it effectively ignores the language context (low $\Delta$Box and $\Delta$Conf) (Table 1). On average, the confidence of the predicted boxes changes only $0.05$ between $q^+$ and $q^-$. One could see the examples in Figure 1. GLIP models make almost identical predictions for two different queries. Such insensitivity to language context makes it infeasible and unreliable to use descriptions to control model predictions.

## 4.2 Setup

In this section, we apply our approach to two vision-language object detection models GLIP [25] and FIBER [7].

**Models.** The visual backbon of GLIP and FIBER is Swin Transformer [30] and the text backbones are BERT [6] for GLIP and RoBERTa [29] for FIBER. Both models use Dynamic Head [4] as the detection architecture. Built upon the two models, we introduce two model variants: **DESCO-GLIP** and **DESCO-FIBER**.

**Datasets.** Following GLIP [25], we train the models on 1) O365 (Objects365 [39]), consisting of 0.66M images and 365 categories; 2) GoldG that is curated by MDETR [17] and contains 0.8M human-annotated images sourced from Flickr30k [35], Visual Genome [19], and GQA [15]; 3) CC3M [40]: the web-scraped Conceptual Captions dataset with the same pseudo-boxes used by GLIP. We down-sample CC3M to around 1.4M images to save training costs, based on whether high-confidence boxes exist in the image. As illustrated in Section 3, we convert the text caption of each instance into a detailed language description to construct description-rich data.

To evaluate how well the models generalize to novel concepts, we perform a zero-shot evaluation on the LVIS [11] and OmniLabel [38] datasets. LVIS is a popular dataset that has over 1,200 object categories with a challenging long tail of rare objects; OmniLabel is recently proposed and focuses on object detection with diverse and complex object descriptions in a naturally open-vocabulary setting. For evaluation on LVIS, for each category, we append the GPT-3 generated description to the category name; we group several descriptions into one query to save inference time. More details on the evaluation are in the appendix. For OmniLabel evaluation, we follow the original evaluation protocol without modifications. We also verify that the models still possess the ability to perform the conventional detection and grounding tasks as GLIP and FIBER, on COCO [26] and Flickr30K [35]. The evaluation results are in the appendix.

| Row | Model | LVIS MiniVal [17] | | | | OmniLabel COCO [38] | | | | Context Sensitivity | | |
|---|---|---|---|---|---|---|---|---|---|---|---|---|
| | | APr | APc | APf | AP | AP | APc | APd | APd-P | ΔBox | ΔConf | AP |
| 1 | GLIP-T | 20.8 | 21.4 | 31.0 | 26.0 | 18.7 | 45.7 | 11.7 | 31.2 | 0.291 | 0.05 | 4.7 |
| 2 | + Description w/ Entity Name | 20.5 | 23.9 | 35.5 | 29.2 | 23.6 | 47.4 | 14.7 | 36.0 | 0.293 | 0.06 | 5.7 |
| 3 | + Description w/o Entity Name | 25.6 | 25.9 | **35.9** | 30.7 | 24.0 | 46.8 | 16.0 | **37.0** | **0.382** | 0.10 | **10.7** |
| 4 | + Description w/o Name + Neg Cap | **26.5** | **27.1** | 35.8 | **31.3** | **24.7** | **48.2** | **16.6** | 36.2 | 0.381 | **0.10** | 10.5 |

Table 3: Ablation study. Directly appending the description does not improve performance on rare categories (Row 1 v.s. Row 2, LVIS APr). Constructing context-sensitive queries is crucial.

**Implementation details.** We initialize DESCO-GLIP from the GLIP-T checkpoint and DESCO-FIBER from the FIBER-B checkpoint. We fine-tune the models on both the original data and the new description-rich data. For DESCO-GLIP, we fine-tune with a batch size of 16 and a learning rate of $5 \times 10^{-5}$ for 300K steps; for DESCO-FIBER, we fine-tune with a batch size of 8 and a learning rate of $1 \times 10^{-5}$ for 200K steps. Experiments can be replicated with 8 GPUs each with 32GB memories.

## 4.3 Zero-shot transfer to LVIS and OmniLabel

**LVIS.** Our method shows notable improvements over the baselines on the LVIS MiniVal dataset (Table 2). The improvement is particularly prominent for rare object categories (APr), with an increase of 10.0 for GLIP and 9.1 for FIBER. Results on the Val 1.0 set are in the Appendix.

**OmniLabel.** Our method also shows improvements over baselines on the OmniLabel dataset (Table 2). OmniLabel assesses model performance using plain categories (APc), free-form descriptions (APd), and positive descriptions (APd-P). Because our models are trained with description data, they naturally excel in supporting such queries, leading to substantial increases in APd and APd-P compared to the baselines. Specifically, DESCO-GLIP achieves a notable improvement of +4.6, while DESCO-FIBER achieves an even more impressive improvement of +5.0. Furthermore, our model's effectiveness extends beyond free-form descriptions to plain categories as well, as illustrated in the table. This highlights the robustness of our method across different evaluation settings and its ability to achieve improvements in various types of queries. Our method wins the 1st place in the Omnilabel challenge 2023 on all three tracks (see Appendix for details).

## 4.4 Ablation study

In this section, all ablation models are initialized from GLIP-T and trained for 100K steps.

**Directly appending descriptions.** We examine the impact of directly adding language descriptions to text queries, without incorporating context-sensitive query construction. The results are presented in Row 2 of Table 3. The performance on rare categories (APr) sees no improvement but decreases. To further evaluate the sensitivity of the model to contextual changes, we conduct the same context sensitivity analysis as the one described in Section 4.1. The context sensitivity of the model almost remains unchanged (Row 1-2): ΔBox changes only 0.002 and ΔConf changes only 0.01. The results indicate that the model remains as insensitive to context changes as the baseline model. This suggests that the model struggles to accurately interpret and effectively utilize the provided language descriptions when context-sensitive query construction is removed.

**Dropping the entity name.** As in Section 3.2.2, we hypothesize that randomly removing the entity name can force the models to concentrate on the contextual information. Remarkably, the results presented in Table 3 (Row 2-3) demonstrate that this simple and intuitive approach proves to be highly effective. It significantly enhances the model's contextual sensitivity while concurrently improving object detection performance.

| GPT | APr | APc | APf | AP |
|---|---|---|---|---|
| ada | 19.9 | 23.2 | 33.7 | 28.0 |
| babbage | 24.2 | 26.7 | 36.5 | 31.3 |
| curie | 24.7 | 28.4 | 38.2 | 32.8 |
| davinci | 30.8 | 30.5 | 39.0 | 34.6 |

Table 4: Detection performance improves when language model size grows.

**Negative captions.** We also investigate the effectiveness of using language models to generate hard negative

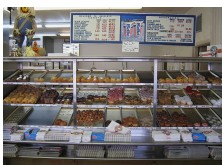 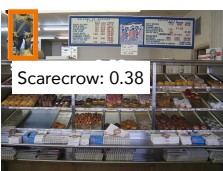 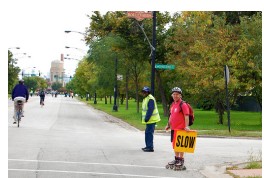 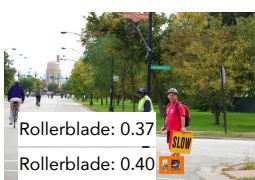

From **GPT-Curie**:
Scarecrow, a kind of object, tall, with a straw in its mouth, could have a hat, could be made of straw.

From **GPT-Davinci**:
Scarecrow, a kind of decoration, made of straw, has a hat and clothes, could have a face.

From **GPT-Curie**:
Rollerblade, a kind of sports equipment, blades that rotate on the ground

From **GPT-Davinci**:
Rollerblade, a kind of sports equipment, wheels attached to a boot, used for skating

Figure 4: Detection performance of DESCO-GLIP improves when given better descriptions. GPT-Curie is a smaller model than GPT-Davinci; it gives less accurate descriptions for objects.

captions. As shown in Row 4 of Table 3, including neg-
ative captions can improve the model detection performance across datasets while preserving its robust contextual comprehension. These results indicate that this technique effectively enhances the model's ability to grasp the subtleties embedded in the given language descriptions.

**Language description quality.** We explore the effect of the language model size on detection performance. We evaluate the pre-trained DESCO-GLIP on LVIS with descriptions generated from the GPT families[4]. As presented in Table 4, higher-quality language models improve object detection performance. This finding highlights the importance of employing strong language models, as they possess the ability to embed valuable visual information through extensive pre-training. We showcase two examples in Figure 4.

# 5 Conclusion and limitations

In this study, we introduced a new paradigm of learning object detection models from language supervision. We show that large language models can be used to generate rich descriptions and the necessity to construct context-sensitive queries. We hope that our method sheds light on empowering vision models with the ability to accept flexible language queries.

While we greatly improve the models' ability to understand flexible language queries, our method has several limitations that can be addressed in future work. 1) We use a large language model to automatically generate the descriptions, which inevitably introduces noise as not all generated descriptions are accurate or beneficial for representation learning. Future work could consider automatically selecting useful descriptions sampled from the language model, similar to [49]. 2) The format of the descriptions we explored is still limited (e.g., "{entity}, a kind of {type}, {list of simple features}"); it might be useful to consider more diverse descriptions by prompting the language model with more diverse prompts. 3) Similar to large language models, querying our model also requires a certain amount of prompt engineering. Future work could explore how to make the model more robust to different kinds of queries.

# Acknowledgement

We would like to thank members of UCLA NLP for their helpful comments. We also thank the reviewers for the valuable reviews. This work was supported in part by DARPA MCS program under contract number N660011924032, DARPA ECOLE program No. HR00112390060, and ONR grant N00014-23-1-2780. We also thank Amazon for AWS credits for computational resources. The views expressed are those of the authors and do not reflect the official policy or position of the Department of Defense or the U.S. Government.

---

[4]https://platform.openai.com/docs/models

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

# A Approach

## A.1 Description generation with large language models

We prompt `davinci-003` with the following prompt:

---

Question: What features should object detection models focus on for a given input? Answer:
Input: **zucchini**, Output: "type": "vegetable", "description": "cylindrical, green, smooth; could have brown and rough stems; could be sliced into round pieces; could have green leaves", "similar objects": ["cucumber", "eggplant", "green bean"]
Input: **zebra**, Output: "type": "animal", "description": "black and white stripes; has a long mane", "similar objects": ["horse", "giraffe", "elephant"]
Input: **apple**, Output: "type": "fruit", "description": "red, round, has a stem", "similar objects": ["orange", "banana", "pear"]
Input: **wok**, Output: "type": "cooking tool", "description": "round, deep, has a handle", "similar objects": ["pan", "pot", "frying pan"]
Input: **ambulance**, Output: "type": "vehicle", "description": "red; has a glaring siren; could with a stretcher", "similar objects": ["police car", "taxi", "garbage truck"]
Input: **lantern**, Output: "type": "lighting tool", "description": "round; could be made of papers", "similar objects": ["lamp", "flashlight", "candle"]
Input: **{entity}**

---

Table 5: Text prompt used to sample descriptions from large language models.

We construct a vocabulary of $10K$ entities by extracting noun phrases from the pre-training text corpus (GoldG and CC3M) using NLTK. As we also use Objects365 for training, we add the categories from Objects365 to the entity vocabulary. We query the language model to generate descriptions for the entities.

## A.2 Context-sensitive query construction

As shown in Table 5, when prompting the language model, we also ask the model to name a few "similar objects". Thus, when constructing a query of descriptions, we include both positive descriptions and several negative descriptions for such "similar objects". GLIP has a max query length of 256 tokens. On average, we can pack 8 descriptions into one query. We randomly drop descriptions if the length exceeds the length limit.

For image caption data, we use to prompt in Table 6 to generate negative captions. Note that while the prompt asks the LLM to generate object descriptions and confusable objects, we find that the prompt in Table 5 gives better results for this purpose; thus we do not use the description and confusable objects given by prompt in Table 6.

## A.3 Query construction

---

**Algorithm 1** Generating Queries for *Detection* Data

---

**Input:** $T, E, V$
1: $Q^- \leftarrow \text{RandSample}(V \setminus E)$
2: $q^* \leftarrow \text{ShuffleConcate}(E \cup Q^-)$
3: $q^*, T \leftarrow \text{LabelAssign}(q^*, T, E)$

---

**Algorithm 2** Generating Queries for *Grounding* Data

---

**Input:** $T, D, C$
1: $Q^- \leftarrow \text{RandSample}(D \setminus \{C\})$
2: $q^* \leftarrow \text{ShuffleConcate}(\{C\} \cup Q^-)$
3: $q^*, T \leftarrow \text{LabelAssign}(q^*, T, C)$

---

Figure 5: Algorithms for generating queries from detection data and grounding data for GLIP. **Algorithm 1**: Compare to DesCo, note that no positive entities are dropped from the query; thus $B$ is not involved in the query construction. **Algorithm 2**: $D$ is the corpus of image captions. Compare to DesCo, note that the positive caption $C$ is always included in the query; this creates the hallucination issue.

Given a caption, perform the following:
1. Extract the visually concrete objects and give a description for what object detection models should focus on for an given input. Avoid common objects such as "person", "dog", "actor", "cat", "man", "woman", "boys", "girls".
2. Name some negative visual objects can be confused and give descriptions. Focus on the visual differences between the positive and negative objects.
3. Write some hard negative captions for a given positive caption, by replacing the nouns, adjectives, and verbs with visually similar but wrong words. Do not replace the nouns that refer to people.

Positive caption: an ambulance is rushing down the street with its lights flashing
1. Visually concrete objects and descriptions: "ambulance": "a kind of car, red light, red and white patterns on the body"
2. Objects that can be confused with the above objects: "police car": "a kind of car, blue and red glarring light, blue and white patterns", "taxi": "a kind of car, yellow, has a notch"
3. Negative captions: ["a police car is rushing down the street with its lights flashing",]

Positive caption: the aircraft carrier is enveloped in fog as it sits in its berth
1. Visually concrete objects and descriptions: "aircraft carrier": "a kind of ship, large, has a long deck, has runways, has airplanes on the deck"
2. Objects that can be confused with the above objects: "cruise ship": "a kind of ship, ususally white, luxury, has swimming pools", "cargo ship": "a kind of ship, has boxes on the deck"
3. Negative captions: ["the shipping boat is enveloped in fog as it sits in its berth", "the cargo ship is enveloped in fog as it sits in its berth"]

Positive caption: the beans in the red silicone cup
1. Visually concrete objects and descriptions: "beans": "a kind of vegetable, small, round, usually green"
2. Objects that can be confused with the above objects: "apples": "a kind of fruit, usually red", "beeds": "a kind of decorations, small, round, colorful"
3. Negative captions: ["the beans in the green silver cup.", "the apples in the red silicone cup.", "the beans in the red porcelain cup."]

Positive caption: bulldozer on a building site 1. Visually concrete objects and descriptions: "bulldozer": "a kind of heavy machinery, has a large blade in front, used for pushing debris or soil"
2. Objects that can be confused with the above objects: "excavator": "a kind of heavy machinery, has a long arm and scoop, used for digging and moving materials", "crane": "a kind of heavy machinery, has a long arm and hook, used for lifting"
3. Negative captions: ["crane on a demolition site", "backhoe on a construction site"]

Positive caption: A girl standing in a shallow creek , wearing stilts .
1. Visually concrete objects and descriptions: "stilts": "a kind of shoes, has two long poles"
2. Objects that can be confused with the above objects: "high heels": "a kind of shoes, has a high heel", "sandels": "a kind of shoes, has a strap, usually worn in summer"
3. Negative captions: ["A girl standing in a shallow creek , wearing sandels", "A girl running in a deep river , wearing stilts"]

Positive caption: {**caption**}

Table 6: Text prompt used to sample negative captions from large language models.

# B  Experiments

## B.1  Context-sensitivity test

To create the test set, we go over the LVIS validation set and for each image that has a rare object (defined by the LVIS taxonomy), we query the model with a $q^+$ (the description of the rare object, without the object name) and a $q^-$ (the description of a confusable object, without the object name). The confusable object comes from prompting the LLM as shown in Table 5. Then we calculate the difference between the predictions made by the model for $q^+$ and $q^-$.

For $q^+$ and $q^-$, the model will give two sets of predictions (two lists of boxes). We first match the two lists of boxes by their IoU overlap. Then $\Delta$Box is the percentage of boxes that have high IoU overlap ($>0.5$) between the two sets of predictions; $\Delta$Score is the confidence score changes for those matched boxes. With higher $\Delta$Box and $\Delta$Score, the model predictions change more drastically. The evaluation data and code will be released upon acceptance.

## B.2 Evaluation on COCO and Flickr30K

| Model | COCO mAP | Flickr30K Val R@1 | R@5 | R@10 |
|---|---|---|---|---|
| GLIP-T* | 44.6 | 85.6 | 95.8 | 97.3 |
| DESCO-GLIP | 45.8 | 85.3 | 95.8 | 97.3 |
| FIBER-B | 49.3 | 87.1 | 96.1 | 97.4 |
| DESCO-FIBER | 48.9 | 86.9 | 96.4 | 98.0 |

Table 7: DESCO-GLIP and DESCO-FIBER maintain similar performance to GLIP and FIBER on common object detection and phrase grounding. DESCO-GLIP and DESCO-FIBER are fine-tuned with a smaller batch size than GLIP and FIBER; thus some minor performance drops are expected.

In Table 7, we report the performance on common object detection (COCO, zero-shot) and Flickr30K. The performance of DESCO-GLIP and DESCO-FIBER is similar to GLIP and FIBER. GLIP-T* is the checkpoint that achieves the best performance on LVIS released in `https://github.com/microsoft/GLIP`; it has slightly worse performance than GLIP-T on COCO. We used GLIP-T* to initialize DESCO-GLIP, thus we compare with GLIP-T* in this case.

## B.3 Evaluation on LVIS

| Model | LVIS Val APr | APc | APf | AP |
|---|---|---|---|---|
| GLIP | 10.1 | 12.5 | 25.5 | 17.2 |
| DESCO-GLIP | 19.6 | 22.0 | 33.6 | 26.2 |
| FIBER | 18.0 | 21.6 | 35.0 | 26.3 |
| DESCO-FIBER | 23.0 | 26.3 | 38.5 | 30.5 |

Table 8: DESCO-GLIP and DESCO-FIBER achieve strong performance on LVIS val 1.0.

For evaluation on LVIS, for each category, we generate descriptions with prompts in Table 6. When doing inference on an image, as the vocabulary is large, concatenating descriptions of all categories into a long prompt results in a extremely long prompt. We adopt the same approach as GLIP and FIBER; we construct multiple prompts where each prompt only contains a few category descriptions; the model is queried multiple times and we aggregate the outputs across the multiple forward passes.

We evaluate models on the full validation set of LVIS using the fixed AP protocol. It can be seen that our approach achieves large performance gains compared to the baselines.

## B.4 OmniLabel Challenge 2023 winning entry

We submit DESCO-FIBER to the OmniLabel Challenge 2023[5] and won the 1st place. Different from the model reported in the main text, the submitted version is first initialized from DESCO-FIBER, then undergoes a second-stage pre-training on Objects365, GoldG, CC3M, and RefCOCO [52] (including RefCOCO, RefCOCO+, RefCOCOg), with a batch size of 8 for another 200K steps.

---

[5]`https://www.omnilabel.org/challenge/challenge-2023`

