# OpenReview forum: "DesCo: Learning Object Recognition with Rich Language Descriptions"
_NeurIPS.cc/2023/Conference — NeurIPS 2023 poster_

### Official Review · Reviewer_9ZAR · 2023-06-28

**Soundness:** 3 good
**Presentation:** 2 fair
**Contribution:** 2 fair
**Rating:** 4
**Confidence:** 3

**Summary:**

The paper describes an approach to open world object detection that generates detailed object descriptions with GPT to improve performance. The main advance of the paper is showing that the removal of the entity name, and just using it's description, forces the network to learn the "context". This is turn leads to better performance on zero-shot tasks. Results are shown on the LVIS and OmniLabel datasets in comparison to GLIP and FIBER.

**Strengths:**

The paper clearly demonstrated that the additional contextual information needed to be presented to the network properly to see a performance boost.

The authors' results are clearly better than FIBER and CLIP.

**Weaknesses:**

The writing of the paper make it difficult to understand the approach. I had to read it multiple times to understand how the models were trained and tested. It's still not 100% clear how all the diverse datasets used for training where converted to description-rich data. The paper states "As illustrated in Section 3, we convert the text caption of each instance into a detailed language description to construct description-rich data.". This should be made more concrete.

The paper doesn't describe the training of CLIP and FIBER. A little more background would help better differentiate the contributions of this paper.

The main novelty of the paper is removing the entity name from the description to force the network to use the contextual description. It is not clear this is of sufficient novelty/interest for NeurIPS.

The authors don't explore other prompts for generating object descriptions. This would have greatly added to the paper.

**Questions:**

Where the COCO images removed from training, similar to CLIP?

When generating hard negatives (Line 194), how can you ensure they're actually negative?

The example of the mallet in the paper states a mallet has a wooden handle, but the image contains a mallet with a red handle (could be made from any material). Is the model really learning a grounding for the context, or just querying with a bag of words?

On line 176, it's stated the "mutual information still stays zero". Is this true? The mutual information may be low, but is it really zero?

**Limitations:**

The paper does not provide potential negative societal impact of their work.

---

> ### Author Rebuttal · Authors · 2023-08-10
>
> 1. More details on the training and testing procedure.
>
> Thank you for raising the issue and we will include more details in the next version.
>
> We have included the detailed data generation process in the attached PDF (Figure 1) and we briefly introduce the process in the following. DesCo is trained with two kinds of data: <image, entity> data (the typical object detection data) and <image, caption> data. For the latter, we randomly choose one of two ways to augment it:
> Extract the entities and append descriptions to it. We will also sample “confusable” descriptions for other entities and optionally drop the entity from the query.
> We add hard negative captions generated by a language model; the original caption may be dropped from the query.
> For the former, we just conduct Augmentation A.
>
> &nbsp;
>
> 2. "Describe the training of CLIP and FIBER"
>
> Thank you for the suggestion. We will add more descriptions on the training recipe of the baseline models. For FIBER, the detection training stage is the same as GLIP thus we skip it.
>
> &nbsp;
>
> 3. Novelty
>
> Please check the general response.
>
> &nbsp;
>
>
> 4. "Explore other prompts"
>
> Thank you for the suggestion and we believe it would be an interesting direction to explore. For this paper, as a first step towards complex language-based detection, we focus on proposing techniques to reduce the bag-of-words/hallucination problem and opted to use a rather general prompt/description style. Thus, we left the exploration of different prompts to future work; the fact that our approach works without extensive prompt engineering also speaks to its effectiveness.
>
> &nbsp;
>
> Questions:
>
> - COCO images are removed from training. The pre-training source data are exactly the same as GLIP-T.
>
> - Indeed the language model could give false-negative captions; but manual inspection of the data suggests that this is rare. Our empirical results suggest that the method works well despite such noise.
>
> - That’s a great observation! Indeed, even if we train our system on such context-sensitive data, there is
> no guarantee that the model “understands” the context completely. How much the model learns will depend on the “difficulty” of contrastive learning pairs we construct; thus such a phenomenon can be eased if we could prompt the LM to generate descriptions that require deeper contextual understanding.
>
> - Thank you for pointing this out. The original intention is that if the region labels can be **completely** determined solely by the entity name in all cases, then the mutual information is zero. This is a strong assumption and the mutual information for natural grounding data is not strictly zero. We will revise the wording of the statement.

---

### Official Review · Reviewer_xbf9 · 2023-06-29

**Soundness:** 3 good
**Presentation:** 2 fair
**Contribution:** 2 fair
**Rating:** 5
**Confidence:** 5

**Summary:**

This paper is about open-vocabulary object detection, in which a model is given an image and a sequence of object descriptions and has to predict bounding boxes matching the descriptions. While open-vocabulary detectors can in principle take arbitrarily detailed object descriptions as input, they tend to focus on the main noun in the description and ignore further details in the description. This is problematic at inference time, where the model might be prompted with descriptions of objects that it has not seen during training, and where leveraging additional detail in the description becomes necessary to solve the task.

The present paper addresses this issue by augmenting the training data with detailed object descriptions and then forces the model to use these descriptions.

First, standard object detection datasets are augmented with detailed object descriptions by querying a large language model (GPT-3) to provide additional details about object categories occurring in the detection datasets. The language models are also used to generate hard negative descriptions for similar object categories.

Second, open-vocabulary object detectors are trained using these descriptions. By combining positives with easily confusable negative descriptions, the model is forced to attend to the details in the description. A crucial detail is that the description should not explicitly name the target object type, such that the model is forced to use the description to solve the task.

Using two off-the-shelf open-vocabulary object detectors (GLIP and FIBER), the experiments show that the proposed data augmentation improves performance on the LVIS and OmniLabel benchmarks.

Ablations show that the biggest boost in performance comes from adding detailed descriptions that do not explicitly name the target entity to the training data. Hard negatives provide a small additional boost.


**Strengths:**

1. Natural language supervision is an important trend. Therefore, general ideas for how to clean or augment natural language to provide better supervision are valuable. The idea to remove the entity name from descriptions goes in that direction.
2. The analysis of how detection performance is affected by the LLM used for data generation (Table 4) is significant because it shows that language quality is an important determinant for downstream task performance. This is useful guidance for other works using LLMs for data generation.


**Weaknesses:**

1. The contributions are insufficient for a NeurIPS paper. The primary contribution is to apply the Winograd format to query construction for object detection. While it is useful to know that this helps in the case of object detection, the idea has been employed extensively in related contexts. Further, using an LLM to generate the object descriptions is not well motivated and not necessary at the scale explored in the paper. Datasets of object descriptions of the size used here (~10000 entities) already exist; it’s not necessary to use an LLM for that. There are many interesting questions that could be explored to make the contributions more substantial. For example, does performance improve when scaling up to larger numbers of object descriptions, which is possible with LLMs? Does the LLM prompt matter? Could similar results be achieved with existing datasets by simply masking the entity name?

2. The paper lacks detail in the methods and experiments. The proposed approaches could not be reproduced from the given descriptions. For example, much more detail is needed about the generation of negative captions. The paper briefly mentions that negative captions are generated using a multi-stage process that involves first prompting a language model, then training a second language model to generate more data. But no details are given about how the first LLM is prompted or how the second one is trained. Please describe this process in sufficient detail to reproduce it.

3. The paper makes some misleading claims. For example, L279 claims that the proposed method “surpasses supervised models by a substantial margin”. The models labeled as “supervised” in Table 1 are very far from the state of the art of supervised LVIS detectors, which reach around 60 AP and 50 APr. This is therefore a meaningless and misleading comparison.


**Questions:**

Questions

1. LVIS evaluation (Appendix B.3): What exactly do you mean by “fixed AP protocol”? Do you mean the evaluation proposed in https://arxiv.org/abs/2102.01066? That paper makes various recommendations and doesn’t specify a precise protocol. How exactly do you modify the standard LVIS AP metric to “fix” it? Please provide code or a detailed descriptions sufficient for reproduction.

2. Suggestion for improving the paper: Build on the unique possibilities afforded by LLMs and study those more. Scale up, study prompt choices, study description diversity etc.

Typos

* Typo in abstract: OminiLabel.
* L92 missing word: “…pretrains models large-scale image-caption pairs to learn transferrable representations.”
* Appendix L37 typo: “protocal”

**Limitations:**

The paper does not discuss limitations.

---

> ### Author Rebuttal · Authors · 2023-08-10
>
> Thank you for your detailed review.
>
> 1. “Insufficient Contributions”
>
> We list our core contributions in the general response, and we would like to emphasize the following points.
>
> As Reviewer Wg2t pointed out, DesCo is the first approach to support complex-language based detection, while “recent works in open-vocabulary object detection still focus on mining region-word (noun) correspondence”. Our paper solves **an important and timely problem** of how to improve detection systems with language instructions. This problem is featured in the recent OmniLabel Challenge 2023 at CVPR and our system achieves the best performance. As we showed in the ablation study, the improvement is not only from engineering efforts but the novel design of our training algorithm.
>
> While there are some arxiv/contemporary papers that touch on ideas of using hard negative examples (as noted by Reviewer BSUS), they mostly focus on image classification. We study a more difficult problem (detection/dense prediction).  The model is essentially asked to perform classification over thousands of potential regions for one image (in contrast to outputting one label per image); it is thus more prone to output false-positive boxes for negative queries. And our approach successfully reduces such hallucinations.
>
> We are happy to give a more in-depth discussion and comparison if the reviewer could give pointers to relevant literature.
>
> &nbsp;
>
> 2. "Using LLMs is not necessary"
>
> LLM is an easy and scalable tool to obtain training descriptions of desired style. One relevant large-scale “description” dataset is WordNet, which does not include detailed visual specifications. For example, for the concept “eclair”(Figure 1 top-left), the WordNet definition is “oblong cream puff” while LLM gives “a kind of food, long, cylindrical pastry, filled with cream, topped with chocolate”. In addition, the 10K entities are automatically extracted from the grounding+CC3M datasets; many of the entities are multi-word phrases (e.g., “stop sign”) which do not have definitions in WordNet. Any additional pointers to related datasets are welcome and we will cite and discuss them in the paper.
>
> LLM also allows easy transfer to downstream tasks where we can query any novel test concepts on the fly. In fact, for the main evaluation benchmark LVIS, we do not explicitly make sure the evaluation categories are included in the pre-training description corpus; rather, by word matching, only 699 out of 1200 LVIS categories appear in our 10K entity list; this again shows the flexibility our approach offers. Thus, replacing the LLM in our paper with existing hand-labeled resources would involve significant efforts in data curation and taxonomy mapping.
>
> &nbsp;
>
> 3. Detail in the methods and experiments
>
> Thank you for the suggestions. We have added the prompt used to generate the negative captions (Table 1) as well as the negative augmentation algorithm (Figure 1) in the attached PDF file. We will also open-source the code for reproducing the experiments.
>
> &nbsp;
>
> 4. Misleading claims
>
> Thank you for the comment. We will remove this claim and re-exam other claims.
>
> &nbsp;
>
> Response to questions:
>
> 1. Thank you for checking the cited paper. The fixed AP metric is included into LVIS Challenge 2021 as an official metric (https://www.lvisdataset.org/challenge_2021) and we used the implementation provided by Dave et al. (master). The evaluation metric is also adopted by MDETR, GLIP, FIBER, DetCLIP. We will make the evaluation protocol more clear in the next version.
>
> 2. Thank you for suggesting these interesting directions. As a first paper working on complex language-based detection, we focus on reducing the bag-of-words/hallucination problem and opted to use a rather general prompt/description style, following Menon et al. Thus, we left the exploration of different prompts to future work; the fact that our approach works without extensive prompt engineering also speaks to its effectiveness.
>
> Menon & Vondrick, Visual Classification via Description from Large Language Models, ICLR 2023.

---

> > ### Comment · Reviewer_xbf9 · 2023-08-15
> >
> > Thanks for the response. The response addresses my questions. I still think that the contribution is borderline, but raised my recommendation to borderline accept in light of the response (e.g. motivation for using LLMs) and the arguments of the other reviewers.

---

### Official Review · Reviewer_hkVU · 2023-07-03

**Soundness:** 3 good
**Presentation:** 3 good
**Contribution:** 3 good
**Rating:** 6
**Confidence:** 5

**Summary:**

This paper leverages rich descriptions generated by LLMs to encourage GLIP/FIBER to pay more attention to object attributes aside from object names when performing detection tasks. A more robust model is resulted from training on object-description alignment as the model is able to leverage attributes for recognition as well as avoiding confusable objects.

Experiments show that key to the success of this method is the inclusion of hard negative descriptions without object names, in order to effectively motivate the model to use the information in addition to object names. By doing so, the model also overcomes one of the previous limitations where a detection model is always predicting objects even though the description mentions non-existing objects in the given image.

**Strengths:**

1. This paper leverages hard negatives to force a VL model to learn the object-attributes alignment in addition to merely mastering object-noun alignment. This brings significant improvement in zero-shot recognition of long-tail objects.
2. This paper enriches captions in existing image-text datasets with large language models. Those enriched text descriptions can be used to train better language-aware vision models.
3. Overall, this paper demonstrates a viable path of distilling the knowledge present in LLMs into better object grounding abilities in VL models.

**Weaknesses:**

1. There lacks a test for the false-positive rate. The proposed training setup with carefully hard negatives is alleged to address the issue mentioned in line 181-184. However, there lacks evaluation on how much improvement there is with respect to hallucinating entities that are mentioned by the description but don't exist in the image.
2. There lacks a test for how well the model is able to "reject" a grounding attempt based on misaligned attributes. For example, when the image contains a red umbrella, Line 195-196 ("Note that the label of the word mallet is affacted by the context") suggests that the expected behavior is to ground "red umbrella" while not predicting any grounding if the text says "yellow umbrella", even though the entitiy name is correct. It'd be helpful to evaluate how well the model captures the nuances exemplified by these two cases.
3. Lack of error analysis. For example, in what cases do the relative confidence scores for Q+ and Q- flip after Desco training?
4. As shown in Figure 3, although the descriptions might contain multiple entity names, it seems that the target entity is always the first token, while entities appearing later in the description are supposed to be "modifiers". Would this bring an undesirable bias? How well does the model perform if the target entity appears in a different location in the description?

**Questions:**

In Table 3, what is APd-P as apposed to APd?

**Limitations:**

More experiments and analyses could be done to better justify the proposed training method.

---

> ### Author Rebuttal · Authors · 2023-08-10
>
> Thank you for the detailed review and constructive comments.
>
> **1. False-positive rate / reject grounding attempt / error analysis**
>
> &nbsp;
>
> A. The main metric AP penalizes false positives
>
> While we did not explicitly include a false positive metric, the AP metric we used already penalizes false positives. During evaluation, we will feed the model **both positive and negative** queries; the model will predict boxes and their alignment scores to all queries; we draw the precision-recall curve considering boxes for both positive and negative queries and then calculate Average Precision (AP).
>
> The model will be penalized on AP if it does not successfully reject negative queries. If it outputs a lot of high-confidence boxes for negative queries, such false-positive boxes will out-rank other true-positive boxes and this would bring down its precision at a given recall level.
>
> &nbsp;
>
> B. Difficulty of designing a comprehensive “flip rate” metric
>
> We also considered designing a “flip rate” metric (how often “the relative confidence scores for Q+ and Q- flip”) during development of the project but we find that this will lead us back to the classical AP metric.
>
> To elaborate, we find it hard to define a clear-cut flip-rate metric; for one image, the model will output many bounding boxes for both Q+ and Q- and there are often multiple ground-truth bounding boxes; it is hard to decide when the predictions are flipped when we have to consider the IoU matching between predicted boxes and ground truth boxes.
>
> In addition, oftentimes, the system improves by reducing its confidence for Q- as well as suppressing wrong boxes for Q+, which is hard to capture via a “flip rate” metric. For example, in Figure 1 top-left, the GLIP model gives Q+ (“eclair”) 0.58 confidence and Q- (“tart”) 0.57, while our system gives Q+ 0.37 while completely suppressing Q-. However, this does not constitute a “flip” as for GLIP, conf_Q+ > conf_Q-. For another instance, in Figure 1 bottom-left, GLIP gives multiple wrong boxes for Q+ while DesCo does not; such improvement is also hard to evaluate using a “flip-rate” metric. But the AP metric will capture and reward such improvement.
>
> Based on the above reasons, we decided to use the standard AP metrics. We add two new metrics in Section 4.1, \delta box and \delta conf, which measures how much the prediction changes between Q+ and Q-; we believe that combining these with the AP metric can show that DesCo indeed enables context-conditioned detection.
>
> We are happy to discuss the metric design if the reviewer has specific recommendations. We can also add more qualitative examples to showcase such “flip” in the next version.
>
> &nbsp;
>
> &nbsp;
>
> **2. “The target entity is always the first token”**
>
> Thank you for pointing this out. Indeed this could bring a bias towards the first entity token. We believe this could be mitigated by including more diverse formats of descriptions (e.g., where the target entity appears in a different location), which we leave for future work.
>
> &nbsp;
>
> Question
>
> The APd-P metric is introduced in the OmniLabel paper that only feeds the model positive queries (no negative queries). Thus the model will not be penalized from false-positives on negative queries.
> In fact, in Table 4, comparing line 4 and line 3, the APd-P drops slightly but APd increases; this means that with hard negatives, the model more effectively rejects negative queries.

---

### Official Review · Reviewer_BSUS · 2023-07-06

**Soundness:** 4 excellent
**Presentation:** 3 good
**Contribution:** 3 good
**Rating:** 8
**Confidence:** 4

**Summary:**

The paper tackles the problem of open-world (language-based) object detection and proposes two main things. First, it uses an LLM to obtain detailed descriptions of the object classes of interests and, secondly, proposes a training methodology that forces the network to pay attention to such descriptions through the use of (language) hard negatives.

**Strengths:**

Good paper overall, ticks all boxes. It uses some convincing advanced baselines in a very interesting and current setting, identifies shortcomings, proposes a reasonable solution, and shows the effectiveness well.

**Weaknesses:**

Nothing important to say. As a minor comment, it is a bit hard to be sure about the setting the paper is tackling. The title talks about "object recognition", but the problem is object detection. Then it mentions grounding too. And it is not clear where the bounding boxes come from. Are they proposed by the network? are they given? (one could go to the glip paper, but better be self-contained).

**Questions:**


I have no qualms with the paper as is. A couple minor things the authors might (or might not) find useful:

- I like the results split for "rare" vs "frequent" objects, I think it is quite informative, but it is just mentioned in passing. Adding this split in a table and be more complete might be useful? More generally, some insight into what type of categories benefit the most (as in, finer-grained? ambiguous?...) and which ones don't would improve the paper.

- the hard negatives technique and the explanations are a hot topic so there's several recent papers. I won't add many arxiv papers, but here's two the authors might find interesting, exploring somewhat similar strategies (though for different problems):
CVPR'23 - Teaching Structured Vision & Language Concepts to Vision & Language Models
from arxiv: Waffling around for Performance: Visual Classification with Random Words and Broad Concepts

**Limitations:**

yes

---

> ### Author Rebuttal · Authors · 2023-08-10
>
> Thank you for the review!
>
> 1. Clarification on the setting
>
> Thank you for the suggestion!  We used “object recognition” as an umbrella term because we believe the proposed technique can be applied to other vision tasks as well.
>
> Indeed the specific experiment setting is object detection. We followed the GLIP approach so grounding and detection are effectively the same task – locating entities mentioned in a text prompt; thus we abused the terminology. The bounding boxes are predicted by the network; it is an end-to-end detection system without relying on pre-computed boxes.
>
> We will revise the paper to make the settings clearer.
>
> 2. "Rare" vs "frequent" objects
>
> Thank you for the suggestion. The rare/common/frequent split is following the definition in the LVIS dataset. The APr/APc/APf refer to AP for the rare/common/frequent categories.
>
> It might be hard to completely separate the improvement for rare and common categories as they can be interlinked. We observe that when our model learns to detect rare objects better, it is also less likely to mistake some rare objects as other confusable common objects; thus APc and APf will also improve with better rare object recognition ability due to less false-positives for the common/frequent categories.
>
> 3. Thank you for the references! We will update the references to include these papers.

---

> > ### Comment · Reviewer_BSUS · 2023-08-18
> > **response to the authors**
> >
> > My opinion was already positive to begin with. I've read the other reviews and the rebuttal and there isn't much to change my mind one way or another. I think the paper should be accepted. It seems useful and reasonable, with good results.

---

### Official Review · Reviewer_Wg2t · 2023-07-06

**Soundness:** 3 good
**Presentation:** 3 good
**Contribution:** 3 good
**Rating:** 7
**Confidence:** 3

**Summary:**

This work revisits a key aspect of open-vocabulary object detection: what makes for the class names? It shows that GLIP, the state-of-the-art vision-language model for open vocabulary object detection, behaves like bag of words and could not effectively understand text descriptions of the objects to detect. To tackle this challenge, it turns to LLMs to extend the class names to detailed descriptions to solve the label scarcity issue. It also creates hard negative descriptions to ensure the model focuses on the context. Experiments on two challenging benchmarks (LVIS & OminiLabel) shows strong results and validates the effectiveness of this work.

**Strengths:**

*Originality*: Though the high-level idea of extending the understanding of class names with LLMs shares similarity with [a], the application of this work is fairly original in this field.

*Quality*: This work is well-motivated via re-evaluation of GLIP. The idea is practically simple and clear, and its effectiveness is well-supported by strong performance on two challenging benchmarks. Overall it is insightful and well-shaped for acceptance.

*Clarity*: The delivery is fairly smooth and clear, and I find no major issues in understanding this paper.

*Significance*: Recent works in open-vocabulary object detection still focus on mining region-word (noun) correspondence. I believe the message/intuition from this paper could be helpful for the community.

*Reproducibility*: Code & data release could be very helpful. Release upon acceptance is promised.

[a] Menon & Vondrick, Visual Classification via Description from Large Language Models, ICLR 2023.

**Weaknesses:**

I did not find major concerns in this work. See next section for questions.

**Questions:**

- More examples, especially failure cases of the LLM-generated descriptions could better help understanding the model.
- Currently, the text seems to be more like a general description in terms of representative visual attributes (low-level) and semantic/meaning of the object (higher-level). One interesting direction is to create more diverse descriptions in different aspects (lower/higher-level), and diagnosis the contribution of each. Another question is how well could such fixed text descriptions cope with diverse (or OOD) visual appearances.

**Limitations:**

limitations are discussed in the last section

---

> ### Author Rebuttal · Authors · 2023-08-10
>
> Thank you for the view!
>
> 1. Failure cases of the LLM-generated descriptions
>
> Thank you for the suggestion! In Figure 4, we included failure cases of the smaller LM; but indeed even the largest LM can generate inaccurate descriptions. We will include examples in the next version.
>
> 2. More diverse descriptions
>
> Thank you for this valuable suggestion! As a first paper on this topic, we opted for a rather simple description style and left the exploration of different description styles to future work.
>
> 3. How well could fixed text descriptions cope with diverse (or OOD) visual appearances?
>
> Indeed objects will have various appearances that are hard to summarize / OOD for fixed descriptions. Our assumption is that during training, the descriptions given by LLM should align well with the objects in most cases and the model learns the correct region-word correspondence; during test time, users could hand-craft OOD queries for OOD appearances when in need. E.g., “Detect a chair with three legs”.

---

> > ### Comment · Reviewer_Wg2t · 2023-08-15
> >
> > Thanks to the authors for the response. I have no further concerns and after reading the discussions with other reviewers, I confirm my positive recommendation. In the next version, I expect the authors to consolidate the discussions with reviewers, provide detailed implementation details (release the descriptions provided by GPT3 & the code). I also came across one closely related paper that was made public after NeurIPS submission, and I suggest for discussion in the paper: Multi-Modal Classifiers for Open-Vocabulary Object Detection, arXiv:2306.05493.
> >
> > Some comments:
> >
> > I do not think the standard of NeurIPS is naively tons of new techniques, but also new inspirations/ideas/messages supported with solid presentation, experiment, and analysis. I believe this work could be inspiring to the community. I am also aware of the recent trend of scaling up models and mining aligned supervision from diverse VL data in OVOD, but that is not the only answer and I feel happy to see the promotion of an orthogonal effort.

---

### Author Rebuttal · Authors · 2023-08-10

We thank all the reviewers for the constructive reviews and suggestions! We are encouraged that the reviewers find our paper “**insightful and well-shaped for acceptance**” (Reviewer Wg2t); “demonstrates a viable path of distilling the knowledge present in LLMs into better object grounding abilities in VL models” (Reviewer hkVU);  we compare with “convincing advanced baselines in a **very interesting and current setting**” (Reviewer BSUS).

Below we would like to summarize the key contributions of our paper:

(1). Enabling VL models to understand complex language queries is an important yet challenging task; we point out an overlooked issue of prior attempts: simply presenting descriptions at training time does not encourage the model to utilize them.

(2). We propose a novel approach of generating detailed descriptions and construct Winograd-style training examples using large language models. Our idea of utilizing natural language supervision from LLM for multimodal learning will “be helpful for the community” (Reviewer Wg2t).

(3). DesCo is the first approach to support complex-language-based object detection and shows significant performance improvement on challenging benchmarks such as LVIS, while previous work is still restricted to “mining region-word (noun) correspondence” (Reviewer Wg2t) or focusing on simpler tasks such as image classification. Our paper solves an important and timely problem of how to improve detection systems with language instructions. This problem is featured in the recent OmniLabel Challenge 2023 at CVPR. Our system achieves the best performance on the leaderboard.

---

### Decision · Program_Chairs · 2023-09-21

**Decision:**

Accept (poster)

**Comment:**

This paper proposes DesCo, a novel method that supports complex-language based object detection. It's well-written and motivated, and provides extensive experiments on several datasets, showing that DesCo outperforms existing methods on object recognition accuracy, generalization to unseen domains. The paper also provides qualitative examples of the generated descriptions, demonstrating the interpretability of the model. Most reviewers showed positive feedback to this paper. Although they raised some question as well, most of the questions are not about the novelty or the technical problem of the paper, but about how to make this work stronger and more solid. The authors have also addressed most of the reviewer comments in their rebuttal and promised to incorporate the feedback in the final version. Therefore, I recommend to ACCPET to this paper as a strong contribution to the field of object recognition and natural language processing.